# Cold-Water Immersion and Sports Massage Can Improve Pain Sensation but Not Functionality in Athletes with Delayed Onset Muscle Soreness

**DOI:** 10.3390/healthcare10122449

**Published:** 2022-12-05

**Authors:** Pavlos Angelopoulos, Anastasios Diakoronas, Dimitrios Panagiotopoulos, Maria Tsekoura, Panagiota Xaplanteri, Dimitra Koumoundourou, Farzaneh Saki, Evdokia Billis, Elias Tsepis, Konstantinos Fousekis

**Affiliations:** 1Therapeutic Exercise and Sports Performance Lab, Physical Therapy Department, University of Patras, 26504 Rio, Greece; 2Department of Microbiology, University General Hospital of Patras, 26504 Patras, Greece; 3Department of Pathology, University General Hospital of Patras, 26504 Patras, Greece; 4Faculty of Sports Sciences, Bu-Ali Sina University, Hamedan 65178-38695, Iran

**Keywords:** cold-water immersion, sports massage, delayed-onset muscle soreness

## Abstract

This study aimed to investigate the effects of cold-water immersion (CWI) and sports massage on delayed-onset muscle soreness (DOMS) in amateur athletes. Sixty male amateur athletes were randomised into four equal groups (*n* = 15) receiving either CWI, sports massage, their combination, or served as controls after applying plyometric training to their lower extremities. The main outcomes measures were pain, exertion, rectus femoris perimeter, knee flexion range of motion, knee extensors isometric strength and serum creatine phosphokinase (CPK) levels examined before the plyometric training, immediately after the treatment, and 24, 48 and 72 h post exercise. We observed no significant differences between study groups in the most tested variables. CWI improved pain compared to the combined application of CWI and sports massage, and the control group both on the second and third day post exercise. Sports massage combined with CWI also led to a significant reduction in pain sensation compared to the control group. In conclusion the treatment interventions used were effective in reducing pain but were unable to affect other important adaptations of DOMS. Based on the above, sports scientists should reconsider the wide use of these interventions as a recovery strategy for athletes with DOMS.

## 1. Introduction

Delayed-onset muscle soreness (DOMS) is the most common clinical manifestation after high-volume plyometric training for both professional and amateur athletes [1,2,3]. This clinical sign is accompanied by stiffness and a sensation of diffuse pain around the involved joints and muscles [4], as well as negative functional capacity adaptations, such as decreased strength, endurance, flexibility, and neuromuscular control [5,6]. These adaptations usually begin 24 h after exercise and remain for three to four days [3,7]. Despite extensive research aimed at determining the etiology and effective treatment of DOMS, there is currently no clear scientific proof.

DOMS has been associated with the alteration of normal lineage myofilaments, thickening or complete disruption of the Z lines of sarcomeres [3], increased release of muscle cell enzymes (creatine kinase [CK]) because of injury during the first one to three days after eccentric exercise [8], and swelling as a result of the production of prostaglandin E2 (PG2) [8,9]. Several theories and respective causative factors have been proposed regarding DOMS etiology, including: (a) the accumulation of lactic acid in the muscle [10]; muscle spasms [11]; (c) the disorganisation of sarcomeres [12]; (d) connective tissue damage [13]; (e) an acute inflammatory response [14]; (f) the loss of calcium homeostasis in the injured muscle fibre [10], and (h) the nerve growth factor (NGF) [15]. However, none of these factors have been credited as the dominant one, leading to the conclusion that the etiology of this syndrome is multifactorial and results from a combination of these factors. What is certain is that DOMS is a result of excessive eccentric exercise stress, such as jumping-plyometric exercises or running downhill and cutting activity drills [16,17].

Symptoms of DOMS can range from muscle tenderness to severe debilitating pain. Therefore, it is clinically significant to design therapeutic interventions for these symptoms [3]. Several therapeutic interventions have been proposed for the prevention and rehabilitation of DOMS symptoms [18,19,20,21,22,23,24,25,26]. These include massage therapy, cryotherapy, exercise, compression garments, stretching exercises, electrotherapy, ultrasound, myofascial release techniques, acupuncture, and sports taping. Nevertheless, the most frequently used rehabilitation techniques for the management of DOMS in athletes are massage and cryotherapy in the form of cold-water immersion (CWI), or even their combined application [18,24]. Sports massage has been evaluated mainly on the basis that it contributes to reducing the perception of pain while also reducing the negative effects on the athlete’s functional capacity, although the results of relative studies are not supportive of that belief. Several researchers indicate that sports massage [27] is ineffective in reducing DOMS’ negative adaptations (stiffness, strength reduction, hematologic muscle damage indicators, performance reduction), while others [20,25] reported a positive effect of massage on pain reduction. In addition, the effectiveness of CWI in reducing DOMS symptoms is contradictory, as some studies [28,29] concluded that CWI was effective in treating the dominant negative effects of DOMS, while others [30] did not. Furthermore, no research to date has evaluated the combined application of these two therapeutic techniques on DOMS management.

Based on these contradictory results, it is evident that there is currently no definitive answer concerning the necessity of applying these techniques for the rehabilitation and prevention of DOMS symptoms and manifestations. If the individual application of these commonly used therapeutic interventions could reduce the detrimental effects of delayed muscle soreness in athletes when done with standard procedures, their combined application could lead to better results. Based on the above, the present study aimed to investigate the effects of CWI, sports massage and their combined application on DOMS in amateur athletes.

## 2. Methods 

The present study has been registered at www.isrctn.com following identification number: ISRCTN85372618. (Registered 13 May 2020).

### 2.1. Participants

Sixty male amateur athletes participating in team sports (soccer, handball, volleyball) with an average age of 21.1 years, height of 1.76 m and body weight of 77.55 kg were randomly divided by a third party into four equal groups (*n* = 15) using an online random generator (https://www.randomizer.org/, accessed on 15 October 2022): a CWI group, a massage group, a combined massage and CWI application group, and a control group. All athletes were informed of the procedures of the research and provided written consent for voluntary participation. The participants were not aware of the specific purposes of the study so that their performance would not be affected. The study adhered to CONSORT guidelines and was approved by the University Institutional Review Board. The inclusion criteria included athletes with no serious lower limb injuries in the last six months prior to the study, those who trained at least three times a week, and those who abstained from any kind of training and analgesic means during the research week. Finally, the measurements were made at the non-dominant lower limb of the athletes. 

### 2.2. Outcome Measurements 

The variables evaluated in the present study were based on literature-accepted indicators (Newham et al., 1983 [1]), such as the feeling of fatigue (BORG scale-original version–6–20 scale) [31] and muscle pain (VAS scale), the perimeter of the rectus femoris with and without contraction, the range of motion (ROM) of knee flexion, the maximum isometric strength of the knee extensors, and the level of creatine phosphokinase (CPK) in the blood serum. The evaluation of the above variables was performed before and after the application of a specialised protocol of induced fatigue, and 24, 48, and 72 h after the induced fatigue. CPK was measured before, and 24, 48, and 72 h after exercise.

The feeling of physical fatigue (exertion) was specifically assessed with the BORG scale, while muscular pain was assessed with the VAS scale (0–10 scale). The upper and middle thigh circumference was evaluated using a measuring tape, where the middle thigh point was calculated as the mean distance between the major trochanter and the medial line of the knee, and the upper thigh measurement point was the highest possible evaluation point (gluteal cleft height). Knee flexion ROM was evaluated using a universal goniometer, while knee extension isometric strength (Peak torque of the knee extensor muscles (N * m − 1)) was performed using an isokinetic dynamometry (Biodex System 3, Shirley, New York, NY, USA). Three maximum isometric contractions of five seconds were performed with a seven-second rest between each, and the average of the three attempts was used in the analysis. The perimeter of the rectus femoris (with and without contraction) was calculated using ultrasound diagnosis with a Mini Focus Ultrasound Scanner (ICEN Technology Company Limited, Guangzhou, China), and was performed from the athlete’s sitting position on the dynamometer. The examination techniques and equipment used in the assessment of the athletes have been used in sports research and are valid and reliable.

The rectus femoris circumference evaluation through diagnostic ultrasound was conducted on the lower third of the rectus femoris. To determine this point, the distance from the anterior superior iliac spine to the upper pole of the patella was measured and divided by three. Two shots were taken without rectus femoris contraction, and two with contraction, at a frequency of 12 Hz. Finally, a Cobas Integra 400 plus analyser (Roche, Basel, Switzerland) and a Cobas integra creatine kinase liquid reagent were used to measure CPK. The blood samples were left at room temperature until a clot formed. Then, they were processed for ten minutes at 800 rpm. The supernatant (serum) was immediately separated and microscopically checked for hemolysis. Samples with hemolysis were discarded. The creatine kinase was then measured in vitro in serum.

### 2.3. Delayed-Onset Muscle Soreness Induction Exercise

The plyometric fatigue protocol used in this study was proposed by Nosaka and Miyama [32], and has been used by several researchers [17,33,34]. It included 100 depth jumps (five sets of 20 jumps) from a 60 cm step. There was a two-minute break between each row and a 10-s margin between each jump. In this particular protocol, the athlete, after climbing the ladder, waited for the researcher’s signal to perform deep jumps accompanied by an explosive contraction of the knee extensor muscles and an explosive vertical jump. The main purpose of selecting this fatigue-induced protocol was to simulate the actual conditions of the athletes’ exercises. After completing the fatigue programme, the athletes followed a cool-down programme consisting of 15 min moderate-intense running at 50% of VO_2_max on a treadmill, and static stretching for the knee muscles to simulate the normal training conditions used to decrease the adverse effects of training.

### 2.4. Interventions

The athletes received either CWI (CWI group), sports massage (massage group) or a combined application of sports massage and CWI (combo group) immediately after the completion of the cool-down program. Participants of the fourth group (control group) did not undergo any therapeutic intervention and proceeded to the evaluation immediately after the cool-down program。

The athletes in the CWI group were immersed in a container of iced water at 10 °C for 10 min. The water level reached up to the anterior superior iliac spine [28] and the water temperature was controlled with a liquid thermometer. The athletes in the sports massage group received sports massage for 20 min (10 min for each quadriceps). Sports massage was done slowly but with increased intensity to affect the deep tissues (Law et al. [20]) utilizing strokes such as effleurages, petrissages, compressions, stripping massage strokes and tapotements. These strokes were applied on the entire surface of the quadriceps in the same sequence and duration (2 min each). The athletes of the combined application group were initially given a 20-min massage followed by a 10-min CWI. Participants were instructed to avoid any physical activity that may interfere with the study outcomes for the following days after the intervention All the measurements and therapeutic interventions were performed by expert physiotherapists of the Therapeutic Exercise and Sports Rehabilitation Lab of the University of Patras, who were blinded to the study scope.

### 2.5. Statistical Analysis

To compare the effectiveness of the intervention groups, and to investigate their effects over time, the Two-Way Repeated Measures ANOVA method (Two-Way RM-ANOVA) was used with univariate analysis. The graphs “Normal Q-Q plot” and “Detrended Normal Q-Q plot” were studied to check the condition of normality. The distribution of the dependent variable in each combination of the related groups was found to be approximately normally distributed. For the calculation of sample size, for Two-Way RM-ANOVA analysis we used G-power software (G*Power-Statistical Power Analyzes software for Windows-RRID: SCR_013726) [35]. Based on the data: (a) number of categories = 4; (b) number of measurements = 3; (c) correlation among repeated measures = 0.5; (d) non-sphericity correction epsilon = 1; (e) error type I = 0.05; (f) partial η^2^ = 0.05 (small effect size), and (g) power = 90%, the minimum total sample size was estimated to be 60 (15 for any group). The condition of sphericity was checked by Mauchly’s sphericity test. For statistical analysis of the data, the statistical software SPSS-25 (IBM SPSS for Windows Version 25.0, IBM Corp., Armonk, NY, USA) was used. The minimum value of the statistical significance level (the *p*-value) in all statistical tests was set at 5% (*p* < 0.005).

## 3. Results

The participants’ functional data and symptoms before and after the application of the fatigue protocol and therapeutic interventions are shown in Table 1.

The analysis of variance for Two-Way RM-ANOVA, with Greenhouse-Geisser correction and the use of post-hoc controls and the Bonferroni correction, showed that the feeling of fatigue (BORG scale), the isometric force of the knee extensions, the knee flexion ROM and circumference of the thigh, and CPK, were all statistically significant, both between the day-1pre and post, day-1pre and day-2, and day-3 and the, day-1pre and day-4 measurements, for all participants (Table 2). In particular, the Borg and perimeter of the rectus femoris relaxed and contracted, and CPK increased significantly after the plyometric fatigue protocol (first to second measurement) and decreased significantly (returned) from day-1pre and day-2 (CPK) or the day-4 measurements (Borg scale, circumference) for all participants. In contrast, the knee extension isometric strength and the knee flexion ROM decreased significantly between the day-1pre and post measurements (post fatigue protocol) and then increased significantly between the day-1pre and day-4 measurements, approaching the initial pre-fatigue values for all participants. No statistically significant differences were observed between the intervention groups for all these variables.

The sensation of pain varied significantly between both Day-1pre to Day-1post and the day-1pre to day-2 measurements for all participants (Figure 1). In all groups, pain increased significantly after the application of the fatigue protocol. However, cryotherapy improved pain sensation compared to the combined application of sports massage and cryotherapy; this was also evident in the control group on the second and third day after the measurements (between the day-1pre and day-2 measurements). The application of sports massage and the combined application of sports massage and cryotherapy led to a significant differentiation (reduction) of the sensation of pain compared to the control group, in which the pain increased linearly between the first and second day. This significant difference between the athletes who received the therapeutic interventions and those who did not (control group) remained throughout all measurements (from the second to the fourth day).

## 4. Discussion

Plyometric exercise makes up an important component of athletic training and performance, and is an efficient method to induce muscle damage. Consistent with previous research, perceived soreness increased significantly following plyometric exercise, peaking between 24 and 48 h before returning toward participants’ baseline values after 72 h [3,7]. The protocol that was used [32] to induce muscle damage was successful, as the indicators of DOMS were significantly affected. A significant increase was observed in creatine kinase activity, the sensation of soreness and pain level, and in the femoral rectal perimeter, with and without contraction. Furthermore, there was a decrease in the range of motion, and isokinetic muscle strength was adversely affected by the plyometric fatigue protocol (first and second measurements).

The findings of the study showed that the application of all three interventions did not lead to a significant reduction in the negative functional effects of eccentric load training compared to the controls. Specifically, the sensation of exhaustion (BORG scale), the isometric force of the knee extensions, the ROM of the knee flexion and thigh circumference and the CPK were statistically significant for all participants, both between the first and second, the first and third, and the first and fourth measurements. In particular, after the plyometric fatigue protocol (day-1 pre and post measurement), the Borg scale and the perimeter of the rectus femoris relaxed with contraction, and CPK increased significantly and decreased significantly (returned) from the day-1 pre to day 1(CPK) and the day-3 measurement (Borg scale, circumference), respectively, for all participants. In contrast, between the first and second measurements (post fatigue protocol), the isometric strength of the knee extension and the knee flexion ROM decreased significantly and then increased significantly between the day-1 pre and day-3 measurements, reaching the initial pre-fatigue values for all participants. For all these variables, no statistically significant differences between the intervention groups were observed. This contrasts with the findings of other research, which have shown positive effects of sports massage and cryotherapy in the form of cold-water immersion to treat the dominant adverse effects of DOMS [20,25,28,29].

The above findings can be partially attributed to the organized cool-down program implemented immediately after the fatigue protocol of the study. Furthermore, moderate to intense running and knee muscle static stretching have been found to significantly reduce the harmful effects of plyometric workouts. This was reinforced by the findings of Gill et al. [36], who reported a significantly faster recovery of creatine kinase activity in interstitial fluid in elite rugby players between the first and fourth days after a rugby match following an active (cycling-based) cool-down compared to a passive cool-down. Furthermore, Wigemaes et al. [37] reported that an active cool-down similar to that applied in the present study, rather than rest recovery, can prevent the initial 0–15 min post-exercise decline in the circulation of white blood cells after strenuous endurance exercise in endurance-trained college-aged males.

The comparison between the therapeutic interventions showed a symmetrical positive contribution in almost all the research variables without a significant difference between them. All interventions helped reduce the negative effects arising after plyometric exercise. When compared with the combined application of sports massage and cryotherapy and the control group, cryotherapy improved pain sensation on both the second and the third day after measurements (between the first and third measurements). The use of sports massage and the combination of sports massage and cryotherapy, respectively, resulted in a significant reduction in pain sensation compared to the control group, in which pain increased linearly between the first and second days. This significant difference in pain sensation persisted in all measurements between the athletes who received clinical interventions and those who did not (control group). These findings agree with the results from other studies [20,25,38], and can be attributed to the adaptation of cryotherapy and sports massage. Specifically, it has been reported that sports massage has positive effects on the negative outcomes of DOMS. Han et al. [25] investigated whether massage was effective in pain relief and gait after DOMS. According to this study, massage on the gastrocnemius for 15 min after DOMS may decrease pain caused by fatigue and inflammation, and pain reduction may lead to a change in gait. Thus, this could be an effective therapeutic mediation that facilitates the recovery of DOMS. Furthermore, Law et al. [20] supported the use of deep-tissue massage to the forearm extensor muscles for six minutes, which can decrease the sensation of stretch pain and hyperalgesia after DOMS induction. Moreover, Davis et al. [38] supported massage as being associated with small but statistically significant improvements in flexibility and pain sensation after induced DOMS. The application of cryotherapy has also been shown to have positive therapeutic effects in treating the symptoms of DOMS. Ingram et al. [22] reported that cryotherapy in the form of cold-water immersion is an effective post-exercise recovery method following exhaustive simulated team sports exercise, underlying lower muscle soreness ratings and improved functional performance (sprints). In the same vein, Ascensão et al. [28] showed that immediately after a one-off soccer match, cold-water immersion reduced muscle damage and discomfort, potentially leading to the faster recovery of neuromuscular function. Additionally, Hohenauer et al. [39] concluded that cryotherapy showed significant effects in reducing the symptoms of DOMS compared to passive control interventions. The positive effects of the combined application of sports massage and CWI in reducing the adverse effects of DOMS can be attributed to the physiological adaptations that follow the alternation of vasodilation and vasoconstriction and the subsequent initial increased and then decreased blood flow to the involved muscles. These alternating physiological adaptations aid muscle repair by reducing swelling and tissue breakdown. Especially after leaving the CWI, the blood vessels vasodilate, allowing fresh blood to move into the muscle. The increase in fresh oxygen helps to remove lactic acid. Cryotherapy is believed to control pain by decreasing nerve conduction velocities.

However, the findings of the present study should be evaluated in light of its significant limitations. First, the sample consisted exclusively of male amateur athletes ranging in age from 18 to 30 years old, thus making it difficult to draw conclusions regarding these responses in professional athletes to these specific recovery techniques. Professional athletes have significant somatic and physiological differentiations-adaptations compared to amateur athletes, and most of them receive the therapeutic interventions of the study systematically and daily. Another limitation to this study was that the variables were only assessed over a 72-h period rather than a longer period, when differences in some measures may have become more evident. Future research should evaluate the effects of their application in professional athletes and with a long-term methodological design. Furthermore, the combined effect of other active (swimming) and passive (compression equipment, cold chambers) interventions should be clarified.

Despite all the above limitations, the present research is characterized by methodological originalities. It is the only research that has applied and evaluated the combination of sports massage and cryotherapy as a recovery strategy for DOMS. Another novelty of this study is the implementation of a structured cool-down program after the fatigue protocol. The importance of this variable lies in simulating real sports conditions, where recovery is an integral part of any type of exercise. This parameter has not been considered by any other study in athletes to date. This organized cool-down program, implemented after the therapeutic interventions, contributed to the results of the present research.

The present findings have significant clinical value, as they support the use of therapeutic methods, cryotherapy, and sports massage to improve the symptoms of DOMS, and the implementation of an organized cool-down program after exhaustive exercise. Regarding effect size, all variables had a large effect and the VAS scale record medium effect at the 4th measurement. It seems that the application of the therapeutic methods of this study (i.e., cryotherapy and sports massage), although they cannot significantly affect athletes’ functional rehabilitation, are capable of reducing DOMS. Based on the above, the present research findings have clinical value if confirmed by corresponding future research. Applying the rehabilitation techniques evaluated in this research to athletes after exhaustive plyometric training can reduce exercise-provoked pain levels and facilitate the application of submaximal training stimuli in the following days.

## 5. Conclusions

An organized cooling down following high volume plyometric exercise may partially lead to a reduced feeling of pain and DOMS combined with CWI and sports massage. The cold-water immersion, massage and the combined massage and cold-water immersion treatment interventions used in this research were successful in minimising pain; however, the other essential parameters of athletes’ functional capacities were not affected. Based on the above, the current study questions the use of these methods by athletes as functional rehabilitation strategies.

## Figures and Tables

**Figure 1 healthcare-10-02449-f001:**
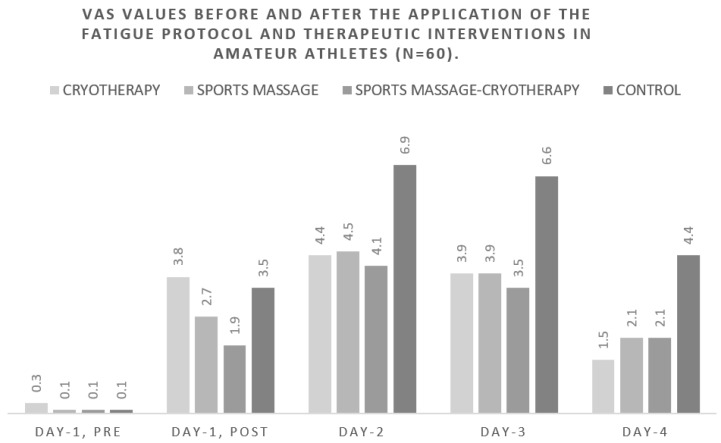
VAS values before and after the application of the fatigue protocol and therapeutic interventions in amateur athletes (*n* = 60).

**Table 1 healthcare-10-02449-t001:** Functional data and symptoms before and after the application of the fatigue protocol and therapeutic interventions in amateur athletes (*n* = 60).

		Cryotherapy (*n* = 15)	Sports Massage (*n* = 15)	Sports Massage-Cryotherapy (*n* = 15)	Control (*n* = 15)
Variables	Measurements	Mean ± SD	Mean ± SD	Mean ± SD	Mean ± SD
BORG	Day-1, pre	7.4 ± 1.0	7.4 ± 1.8	7.3 ± 1.2	7.7 ± 1.6
	Day-1, post	13.1 ± 2.1	11.9 ± 3.1	12.1 ± 2.3	13.9 ± 2.8
	Day-2	11.7 ± 3.6	9.6 ± 3.2	10.7 ± 1.8	10.9 ± 2.5
	Day-3	10.4 ± 2.4	8.5 ± 2.1	9.9 ± 1.9	10.9 ± 3.2
	Day-4	8.1 ± 1.5	7.4 ± 1.6	8.3 ± 1.7	9.3 ± 3.0
VAS	Day-1, pre	0.3 ±0.5	0.1 ± 0.4	0.1 ± 0.4	0.1 ± 0.3
	Day-1, post	3.8 ± 2.1	2.7 ± 1.3	1.9 ± 1.1	3.5 ± 1.8
	Day-2	4.4 ± 2.3	4.5 ± 1.9	4.1 ± 1.7	6.9 ± 1.5
	Day-3	3.9 ± 1.7	3.9 ± 2.1	3.5 ± 1.7	6.6 ± 1.8
	Day-4	1.5 ± 1.0	2.1 ± 1.7	2.1 ± 1.5	4.4 ± 2.0
ROM	Day-1, pre	144.3 ± 8.9	153.1 ± 5.0	151.7 ± 5.1	154.2 ± 4.8
	Day-1, post	141.7 ± 9.5	152.0 ± 5.1	150.2 ± 5.5	153.5 ± 5.0
	Day-2	140.3 ± 9.4	150.9 ± 6.0	149.4 ± 5.8	152.1± 4.6
	Day-3	141.3 ± 9.1	152.0 ± 5.2	150.4 ± 5.7	152.2 ± 5.2
	Day-4	143.7 ± 7.8	153.1 ± 4.6	151.0 ± 5.4	153.3 ± 4.8
STRENGTH	Day-1, pre	289.4 ± 60.4	291.5 ± 58.2	283.7± 57.0	289.0 ± 67.1
	Day-1, post	260.0 ± 52.8	236.1 ± 44.3	238.2 ± 57.7	235.5 ± 70.6
	Day-2	275.7 ± 53.3	256.1 ± 43.3	252.5 ± 67.3	242.5 ± 74.8
	Day-3	293.1 ± 41.9	268.0 ± 41.8	279.0 ± 67.1	257.8 ± 81.1
	Day-4	321.3 ± 51.9	289.6 ± 50.8	287.2 ± 70.0	274.4 ± 83.0
Quadriceps circumference (relaxed)	Day-1, pre	24,819.8 ± 4883.2	25,858.3 ± 4740.6	27,640.6 ± 5159.1	25,783.7 ± 4062.5
	Day-1, post	26,042.2 ± 4588.4	27,825.6 ± 5209.7	29,171.0 ± 5932.2	27,637.8 ± 3469.5
	Day-2	25,412.6 ± 4459.5	26,735.4 ± 5033.8	27,878.7 ± 5215.5	27,323.0 ± 3413.2
	Day-3	25,578.3 ± 5274.2	26,626.8 ± 5859.5	27,583.9 ± 5576.0	27,181.3 ± 3589.0
	Day-4	25,193.4 ± 5640.1	26,658.3 ± 6263.2	27,067.2 ± 5312.6	26,681.0 ± 3864.2
Quadriceps circumference (contracted)	Day-1, pre	24,595.7 ± 6955.5	23,860.2 ± 7769.0	24,377.9 ± 6487.1	23,137.5 ± 6101.3
	Day-1, post	26,132.0 ± 6224.0	25,350.2 ± 7347.0	26,388.2 ± 6814.5	24,287.7 ± 5618.8
	Day-2	25,104.3 ± 6128.0	24,838.2 ± 7609.9	24,928.6 ± 5890.2	24,048.0 ± 5318.5
	Day-3	24,502.2 ± 7244.3	24,038.7 ± 7693.8	24,637.2 ± 6193.5	23,857.7 ± 5586.5
	Day-4	24,034.3 ± 6575.3	23,600.0 ± 7880.7	24,007.9 ± 5816.8	23,434.9 ± 5641.0
CPK	Day-1, pre	638.4 ± 1546.3	263.4 ± 376.9	179.5 ± 71.1	178.9 ± 61.1
	Day-1, post	1713.1 ± 1928.3	1160.6 ± 658.4	1236.2 ± 859.5	874.9 ± 439.4
	Day-2	864.9 ± 756.0	599.8 ± 319.2	683.7 ± 485.2	500.9 ± 346.3
	Day-3	472.1 ± 306.4	447.5 ± 200.1	420.8 ± 292.3	342.5 ± 234.1

**Table 2 healthcare-10-02449-t002:** Test for statistically significant change of research variables by stage and therapeutic intervention.

Variables	Day-1pre–Day-1post Measurement (for All Participants)	Day-1pre–Day-1post Measurement (Comparison between Groups)	Day-1pre–Day-2 Measurement (for All Participants	Day-1pre–Day-3 Measurement (for All Participants-Time Effect)	Day-1pre–Day-3 Measurement (Comparison between Groups)	1st–4th Measurement (Comparison between Groups)
**BORG** **(6–20 scale)**	F (1, 56) = 31.113, *p* < 0.001, η_p_^2^ = 0.357 (95% CI: 0.162, 0.511)	F (3, 56) = 1.011, *p* = 0.395	F (1, 56) = 69.540, *p* < 0.001, η_p_^2^ = 0.554 (90% CI: 0.369, 0.668)	F (3, 56) = 0.567, *p* = 0.639	F (1, 56) = 179.332, *p* < 0.001, η_p_^2^ = 0.762 (90% CI: 0.643, 0.825)	F (3, 56) = 0.534, *p* = 0.661
**VAS** **(0–10 scale)**	F (1, 56) = 71.388, *p* < 0.001, η_p_^2^ = 0.560 (95% CI: 0.377, 0.672)	F (3, 56) = 5.724, *p* = 0.002, η_p_^2^ = 0.235 (95% CI: 0.042, 0.378)	F (1, 56) = 32.444, *p* < 0.001, η_p_^2^ = 0.367 (95% CI: 0.170, 0.518)	F (3, 56) = 5.662, *p* = 0.002, η_p_^2^ = 0.233 (95% CI: 0.041, 0.376)	F (1, 56) = 3.651, *p* = 0.061, η_p_^2^ = 0.061 (95% CI: 0.000, 0.207)	F (3, 56) = 7.688, *p* < 0.001, η_p_^2^ = 0.292 (95% CI: 0.082, 0.433)
**Knee flexion ROM (deg)**	F (1, 56) = 10.624, *p* = 0.002, η_p_^2^ = 0.159 (95% CI: 0.024, 0.326)	F (3, 56) = 0.151, *p* = 0.929	F (1, 56) = 1.129, *p* = 0.293	F (3, 56) = 0.972, *p* = 0.413	F (1, 56) = 9.378, *p* = 0.003, η_p_^2^ = 0.143 (95% CI: 0.017, 0.309)	F (3, 56) = 2.070, *p* = 0.114
**Knee extension isometric strength (Nm)**	F (1, 56) = 14.524, *p* < 0.001, η_p_^2^ = 0.206 (95% CI: 0.048, 0.373)	F (3, 56) = 0.514, *p* = 0.674	F (1, 56) = 54.028, *p* < 0.001, η_p_^2^ = 0.491 (95% CI: 0.296, 0.619)	F (3, 56) = 0.759, *p* = 0.522	F (1, 56) = 119.372, *p* < 0.001, η_p_^2^ = 0.681 (95% CI: 0.530, 0.764)	F (3, 56) = 1.016 *p* = 0.393
**Quadriceps circumference-relaxed (cm)**	F (1, 56) = 21.009, *p* < 0.001, η_p_^2^ = 0.273 (95% CI: 0.092, 0.436)	F (3, 56) = 1.486, *p* = 0.228	F (1, 56) = 15.091, *p* < 0.001, η_p_^2^ = 0.212 (95% CI: 0.052, 380)	F (3, 56) = 1.385, *p* = 0.257	F (1, 56) = 21.704, *p* < 0.001, η_p_^2^ = 0.279 (95% CI: 0.097, 0.442)	F (3, 56) = 1.102, *p* = 0.356
**Quadriceps circumference -contracted (cm)**	F (1, 56) = 13.416, *p* < 0.001, η_p_^2^ = 0.193 (95% CI: 0.041, 0.361)	F (3, 56) = 1.5060, *p* = 0.223	F (1, 56) = 20.205, *p* < 0.001, η_p_^2^ = 0.265 (95% CI: 0.087, 0.429)	F (3, 56) = 1.096, *p* = 0.358	F (1, 56) = 48.002, *p* < 0.001, η_p_^2^ = 0.462 (95% CI: 0.264, 0.596)	F (3, 56) = 1.687, *p* = 0.180
**CPK** **(U/L)**	F (1, 56) = 30.610, *p* < 0.001, η_p_^2^ = 0.353 (95% CI: 0.158, 0.507)	F (3, 56) = 0.273, *p* = 0.845	F (1, 56) = 8.629, *p* = 0.005	F (3, 56) = 0.238, *p* = 0.869	F (1, 56) = 0.964, *p* = 0.331	F (3, 56) = 0.732, *p* = 0.537

VAS = Visual Analogue Scale, ROM = Range of motion, Deg = Degrees, Nm = Newton-metre, cm= Centimetres, CPK= Creatine phosphokinase, U/L = Units per liter.

## Data Availability

The data that support the findings of this study are available from Therapeutic Exercise and Sports Performance Lab, Physical Therapy Department, University of Patras, and are not publicly available.

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
