# Peer review of "Cold-Water Immersion and Sports Massage Can Improve Pain Sensation but Not Functionality in Athletes with Delayed Onset Muscle Soreness"

_healthcare, 2022, doi:10.3390/healthcare10122449_

Round 1

Reviewer 1 Report

To Authors

Thank you very much for giving me a chance to review the interesting paper to investigate the effect of CWI and massage on recovery from plyometric exercise. I assume that the paper is very important for readers in  Helthcare, but I have some major comments before publication.

#1: You are examining the effects of combining CWI and massage. Please explain why you did the two in combination and the order (CWI→massage)you did in the Introduction section.

#2: A detailed description of the method is a bit lacking. Please add a more detailed description of the methods so that other readers can reproduce the same study.

#3: If you investigated the reliability of outcome variables in this study, please add this reliability.

#4: In lines 162-165, what is the basis for the sample size calculation?

#5: In the result section, please explain the interaction effect and the main effect of time. Also, table 2 is hard to understand. I would like it to be easier to understand where the significant differences were.

#6: You mention 1st or 2nd measurement etc, but Day-1 pre or something like that would make the timing clearer. Therefore, please use Day-1 pre and Day 1-post for the table, results, and discussion sections.

#7: In lines 231-240, you stated the active recovery from DOMS but you investigated the “passive recovery” from DOMS, Thus, I don't understand the necessity of this part.

#8: In the discussion section, please discuss the physiological mechanisms by which the combination of CWI and massage elicits a recovery effect from DOMS.

I am looking forward to seeing the revised paper soon.

Author Response

RESPONSES BY THE AUTHORS

#1: You are examining the effects of combining CWI and massage. Please explain why you did the two in combination and the order (CWI→massage)you did in the Introduction section.

Α sentence was added in accordance with the reviewers suggestions

#2: A detailed description of the method is a bit lacking. Please add a more detailed description of the methods so that other readers can reproduce the same study.

We added a sentence that better explains the application of sports massage. we consider that now the methodology is detailed and complete

#3: If you investigated the reliability of outcome variables in this study, please add this reliability.

The present research did not examine the reliability of the measurements

#4: In lines 162-165, what is the basis for the sample size calculation?

For the analysis we have used the G*Power 3, which is a flexible statistical power analysis program for the social, behavioral, and biomedical sciences which has been used in many valid studies. All the

dependent and independent variables of the present research were entered into this system, and the result is what is developed in the text.

#5: In the result section, please explain the interaction effect and the main effect of time. Also, table 2 is hard to understand. I would like it to be easier to understand where the significant differences were.

The effect of time is presented in the table and in the column comparisons of the table 2 involving all participants regardless intervention or non-intervention. This effect is more evident in the comparison from the first to 3rd day.

The referee's thought-proposal about table 2 is correct. The present research, however, is particularly complicated with too many measurements in different periods. Therefore, presenting the research results with many figures and tables would confuse the reader more than it would help him. This is why we created table 2, which presents the statistical significance per measurement period and variable. For the above reasons, we have chosen to emphasize the presentation of the results through the text to make them more understandable to the reader. However, based on the reviewer's suggestion, we created a figure that shows the evolution of the main research variable (pain) during the research course to give a more visual information to the reader.

#6: You mention 1st or 2nd measurement etc, but Day-1 pre or something like that would make the timing clearer. Therefore, please use Day-1 pre and Day 1-post for the table, results, and discussion sections.

The results section (1st measurements etc) were corrected  in accordance with the reviewers suggestions

#7: In lines 231-240, you stated the active recovery from DOMS but you investigated the “passive recovery” from DOMS, Thus, I don't understand the necessity of this part.

This point is an important originality of the study. Specifically, active rehabilitation was applied immediately after the fatigue protocol to simulate real life. In comparison, athletes, after intense plyometric programs, follow typical rehabilitation programs. A sentence was added to the text to explain this paragraph's necessity better.

#8: In the discussion section, please discuss the physiological mechanisms by which the combination of CWI and massage elicits a recovery effect from DOMS.

The following paragraph was added based on the reviewere suggestion “The positive effects of the combined application of sports massage and CWI in reducing the adverse effects of DOMS can be attributed to the physiological adaptations that follow the alternation of vasodilation and vasoconstriction and the subsequent initial increased and then decreased blood flow to the involved muscles. These alternating physiological adaptations aid muscle repair by reducing swelling and tissue breakdown. Especially after leaving the CWI, the blood vessels vasodilate, allowing fresh blood to move into the muscle. The increase in fresh oxygen helps to remove lactic acid. Cryotherapy is believed to control pain by decreasing nerve conduction velocities”

Reviewer 2 Report

Advantages: I can suggest how amateur athletes manage their post-workouts.

weakness: Pre-exercise and post-exercise programs have been studied to reduce DOMS, but combined applications will have a good effect.

1. Please add more clinical significance to manage DOMS in the introduction part.

2. I think the amount of exercise and the muscles used by players in the soccer, handball, and volleyball teams will be different. In future studies, I hope that the subjects will be divided by exercise category.

3. If possible, please present the CONSORT as a picture.

4. Please fill out the reliability and validity of the evaluation tool.

5. Please present the blind method of subjects.

6. Please insert the equipment used in the CWI group

7. If possible, it would be good for readers to present the results in a graph.

8. Please add more clinical significance to management methods for amateur athletes in DOMS in the discussion section.

9. Please check the spelling and dot(.) of the text.

Author Response

. R2 comments

2 1. Please add more clinical significance to manage DOMS in the introduction part.

A sentence was added. Thank you for your comment

2. I think the amount of exercise and the muscles used by players in the soccer, handball, and volleyball teams will be different. In future studies, I hope that the subjects will be divided by exercise category.

Τhis is a valid observation. Different sports lead to varying loads on the athletes' bodies. However, these specific sports require a lot of lower extremity function and involve a lot of plyometric activities such as jumping and sudden direction changes. Neverthelles, for sure, in future studies, subjects will be divided by exercise category.

3. If possible, please present the CONSORT as a picture.

The reviewer's suggestion is correct, but in this case, and since we are given the option of choosing, we will prefer to avoid considering it because it will add extra weight to a text with a lot of information. If the reviewer thinks it is necessary, we will add it

4. Please fill out the reliability and validity of the evaluation tool.

The examination techniques and equipment used in the assessment of the athletes have been used in sports research and are valid and reliable.

5. Please present the blind method of subjects.

The participants were not aware of the specific purposes of the study so that their performance would not be affected. Τhis was clarified in the methods section.

6. Please insert the equipment used in the CWI group

The athletes in the CWI group were immersed in a container of iced water at 10oC for 10 minutes. The water level reached up to the anterior superior iliac spine [28] and the water temperature was controlled with a liquid thermometer Τhis information is contained with details in the method section.

7. If possible, it would be good for readers to present the results in a graph.

Based on the reviewer's suggestion, we created a figure that shows the evolution of the main research variable (pain) during the research course to give a better visual information to the reader.

8. Please add more clinical significance to management methods for amateur athletes in DOMS in the discussion section.

The clinical significance of the findings were added (314-318) in the Discussion section in accordance with the reviewers suggestions

9. Please check the spelling and dot(.) of the text

The text was rechecked for spelling

Round 2

Reviewer 1 Report

Thank you for submitting the revised manuscript.

I have no further additional comments. Congrats!

Author Response

Thanks!